# Acid enhanced zipping effect to densify MWCNT packing for multifunctional MWCNT films with ultra-high electrical conductivity

Hong Wang [1,2,4] ✉, Xu Sun[1,4], Yizhuo Wang[1], Kuncai Li[1], Jing Wang[1], Xu Dai[1], Bin Chen [1,2], Daotong Chong[1,2], Liuyang Zhang [3] & Junjie Yan[1,2]

The outstanding electrical and mechanical properties remain elusive on macroscopic carbon nanotube (CNT) films because of the difficult material process, which limits their wide practical applications. Herein, we report high-performance multifunctional MWCNT films that possess the specific electrical conductivity of metals as well as high strength. These MWCNT films were synthesized by a floating chemical vapor deposition method, purified at high temperature and treated with concentrated HCl, and then densified due to the developed chlorosulfonic acid-enhanced zipping effect. These large scalable films exhibit high electromagnetic interference shielding efficiency, high thermoelectric power factor, and high ampacity because of the densely packed crystalline structure of MWCNTs, which are promising for practical applications.

Carbon nanotubes (CNTs) are comprised of $sp^2$ carbon atoms with tubular structures, like rolled-up graphene sheets[1]. They exhibit remarkable mechanical[2] and electrical properties[3,4] because of the strong C = C bonds and the large π-conjugated system. The combination of these outstanding properties makes them ideal multifunctional materials for applications that require lightweight, flexibility, mechanical strength, and high electrical conductivity[5–8]. However, these outstanding properties have remained elusive in macroscopic films. Making high-performance CNT films introduces major challenges in the material process. Here we report that macroscopic MWCNT films exhibit high electrical conductivity of the high specific electrical conductivity of metals with high strength over 2 GPa, while achieving high electromagnetic (EM) shielding performance superior to previously reported EM shielding materials and high power factor of state-of-the-art inorganic thermoelectric materials, like $Bi_2Te_3$ films[9].

There are two popular ways to make macroscopic free-standing CNT films. One is filtrating the dispersed CNT solution on a membrane (filtrated CNT film)[10]. This approach requires a well-dispersed CNT solution to improve the electrical conductivity of the films[11], which often need the assistance of non-conductive surfactants to de-bundle CNTs, such as sodium dodecyl benzene sulfonate (SDBS), sodium dodecyl sulfonate (SDS), etc. Because good dispersion of CNT solution has been demonstrated to be the key factor that determines the electrical conductivity of the filtrated CNT films[11]. However, filtrated CNT films often have a relatively low electrical conductivity with limited upper boundaries due to the involved non-conductive surfactants[10]. In addition, CNTs are randomly oriented on the filtration membrane in the in-plane direction, which is another reason for the relatively low electrical conductivity in the filtrated CNT films. Because randomly oriented CNTs will lead to the short mean distance for the electron transport according to Mott's theory[12], thus resulting

[1]State Key Laboratory of Multiphase Flow in Power Engineering & Frontier Institute of Science and Technology, Xi'an Jiaotong University, Xi'an 710054, China. [2]School of Energy and Power Engineering, Xi'an Jiaotong University, Xi'an 710054, China. [3]School of Mechanical Engineering, Xi'an Jiaotong University, Xi'an 710054, China. [4]These authors contributed equally: Hong Wang, Xu Sun. ✉e-mail: hong.wang@xjtu.edu.cn

in relatively low electrical conductivity of the filtrated CNT films in the range of 0.0016−0.5 MS/m[13–15]. The alternate CNT film preparation approach is directly winding the synthesized CNT aerogel on a roller (winded CNT film). In this process, CNTs in the films can be partially oriented in the winding direction on the roller and no surfactant is required. Therefore, CNT films prepared by this approach generally have a higher electrical conductivity at the winding direction in the range of 0.02−1.08 MS/m[16]. Yet, these values are still far behind the electrical conductivity of a single CNT in theory, ~100 MS/m[17]. To the best of our knowledge, electrical conductivity >2 MS/m has been seldom reported in macroscopic CNT films. Here we show that exciting properties can be achieved by self-assembling MWCNTs into the crystalline structure in the winded MWCNT films due to the developed chlorosulfonic acid (CSA) enhanced zipping effect.

## Results and discussion

High-quality MWCNT films were synthesized with a floating chemical vapor deposition method and collected by a roller at a winding speed of 3 mm/s (Figs. S1–S3). The average diameter ($D_{CNT}$) of MWCNTs was 2.85 nm, with iron nanoparticles residing both in the core and on the side walls of the MWCNTs (Fig. S1). The length of the MWCNTs/MWCNT bundles was roughly estimated to be about 20 μm as shown in the transmission electron microscope (TEM) images in Fig. S2, which was similar to the values (<20 μm) reported previously[18,19]. The length of the MWCNTs/MWCNT bundles in this work was much larger than the mean free path of 0.27−0.55 μm for MWCNTs reported by ref. [20]. The obtained pristine-MWCNT films (pristine-MWCNT) exhibited isotropic electrical conductivities, which were 0.18 ± 0.03 and 0.04 ± 0.006 MS/m for the directions parallel ($\sigma_\parallel$) and perpendicular ($\sigma_\perp$) to the winding direction (Note that all the properties mentioned in work were measured in the in-plane directions). The size of the pristine-MWCNT films was 25 cm × 28 cm (Fig. S1). The following experiments were mainly focused on the $\sigma_\parallel$ since it was much higher than $\sigma_\perp$.

The pristine-MWCNT films were then purified and densified to increase their $\sigma_\parallel$ as illustrated in Fig. 1a. For the purification process, the pristine-MWCNT films were firstly annealed at 1000 °C under $N_2$ protection to remove the non-conductive amorphous carbon. After that, they were treated with concentrated HCl to remove the metal catalysts and metal oxides. Figure 1b showed that the $\sigma_\parallel$ increased with the annealing time, which reached a platform after 16 h. The maximum $\sigma_\parallel$ of the annealed-MWCNT films (annealed-MWCNTs)

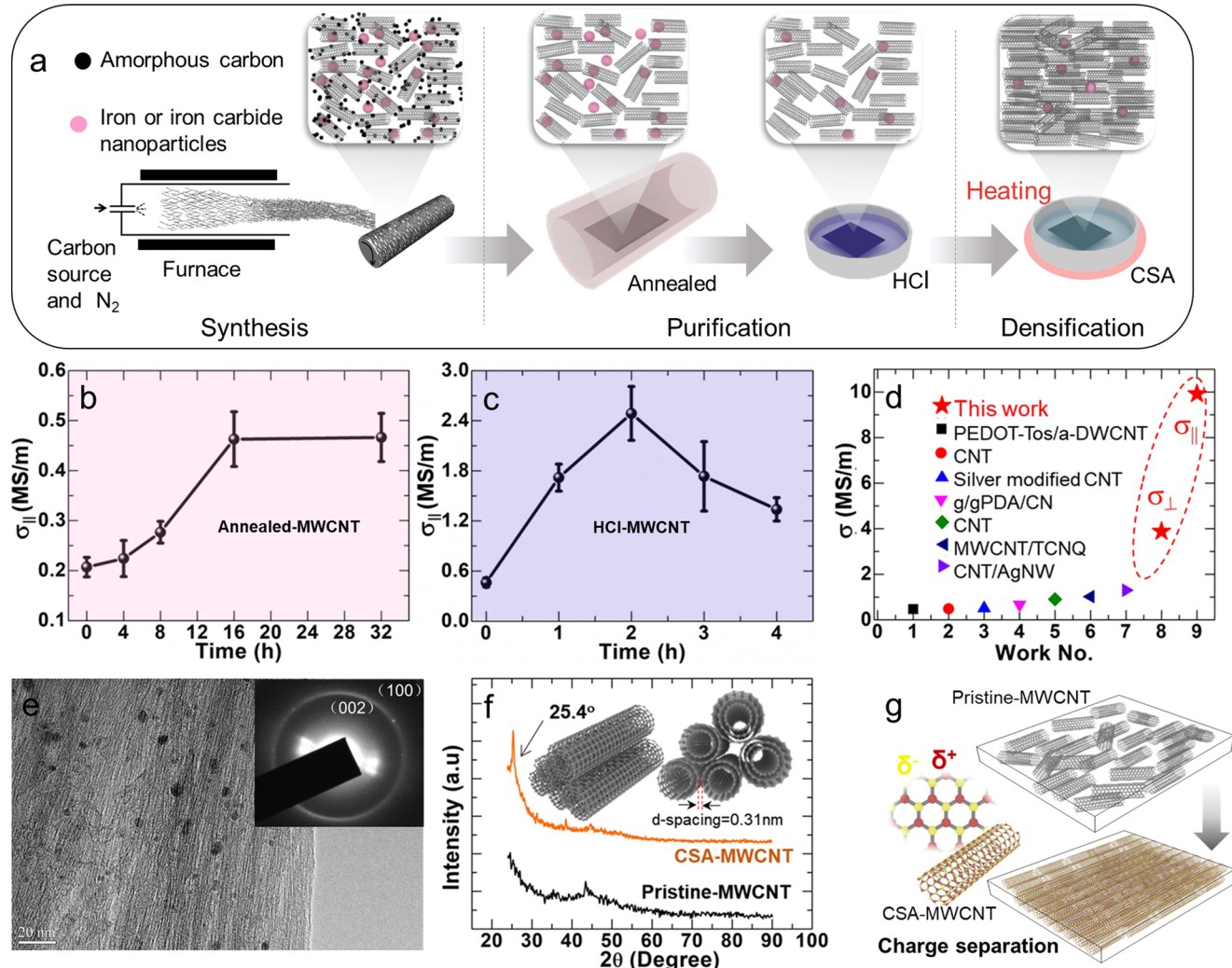

**Fig. 1 | MWCNT film preparation process and basic properties characterization. a** Illustration of the MWCNT film synthesis and treatment processes. **b** The $\sigma_\parallel$ of the MWCNT film as a function of the annealing time, concentrated HCl treating time (**c**). **d** Compared the maximum $\sigma_\parallel$ and $\sigma_\perp$ to the σ of CNT film reported in the literature. **e** TEM image of CSA-MWCNT film with inserted diffraction patterns. **f** XRD spectra of pristine-MWCNT and CSA-MWCNT film. Insert image is the illustration of MWCNT packing in the CSA-MWCNT film. **g** Illustration of densely packed MWCNTs in the films.

was $0.46 \pm 0.05$ MS/m, which was about 2.5 times higher than that of the pristine-MWCNTs. All error bars in this paper were standard deviations. Figure S4 showed the thermogravimetric curves (TGA) of the pristine-MWCNTs and the annealed-MWCNTs under atmospheric conditions. The results indicated that annealing the MWCNT films at high temperatures could remove the non-conductive amorphous carbon, which was the main reason for the increase of $\sigma_\parallel$. Raman spectrum showed that the ratio of $I_G/I_D$ increased from 16.3 to 18.4 after annealing, which also indicated the removal of non-conductive amorphous carbon (Fig. S3).

The annealed-MWCNTs were then treated with concentrated HCl to remove the nanoparticles that were identified to be iron, iron carbide, or iron oxide nanoparticles by X-ray photoelectron spectroscope (Fig. S5). Assuming all the iron was oxidized to be $Fe_2O_3$, the content of iron by mass for the as-synthesized MWCNT film was calculated to be 28.7 wt%. After annealing, the annealed-MWCNT films still contained almost the same amount of $Fe_2O_3$ as shown in Fig. S4, indicating that the amorphous carbon was negligible. Concentrated HCl was commonly used to remove the non-conductive moiety such as iron oxides, amorphous carbon, etc., to improve the electrical conductivity of CNTs[21]. Figure 1c showed that the $\sigma_\parallel$ of concentrated HCl-treated MWCNT films (HCl-MWCNTs) increased with the treating time. The increase of $\sigma_\parallel$ in HCl-MWCNTs was attributed to the reduction of non-conductive contaminants like iron oxides, etc.[22]. After HCl treatment, the content of iron decreased to ~11 wt% because iron without being covered fully by crystalline graphite or MWCNTs has reacted with HCl. The iron was maintained after HCL treatment since the residual iron was all covered by crystalline graphite or embedded into MWCNTs, as demonstrated by the transmission electron microscope images in Fig. S1. The residual iron would contribute to the increase of the electrical conductivity of MWCNT films. It, however, would not boost the electrical conductivity up to a very high level. Similar results have been observed in literature[16,23,24]. In the meanwhile, the HCl doping would also contribute to the increase of $\sigma_\parallel$ in HCl-MWCNTs[25,26]. After reaching a peak value of $2.48 \pm 0.32$ MS/m, the $\sigma_\parallel$ decreased with the rise of treating time because the HCl-MWCNTs films might become loose after being immersed in the concentrated HCl solution for a long time. Figure S6 showed the scanning electron microscope (SEM) images of the morphologies of the pristine-MWCNT, annealed-MWCNT, and HCl-MWCNT. It indicated that the MWCNT films became more and more porous after they were annealed and treated by concentrated HCl due to the leave of amorphous carbon and some iron/iron oxide nanoparticles. The SEM images also showed that HCl-MWCNT films contained much fewer nanoparticles than annealed-MWCNT films.

To densify the MWCNT film for higher $\sigma_\parallel$, a CSA treatment method was developed to crystallize the MWCNT in the film. The thickness of the MWCNT films was identified with SEM images shown in Fig. S7 and Table S1. Densifying the MWCNT films was demonstrated to be an effective way to improve their electrical conductivity since the electrical conductivity would increase exponentially with the reduction of the volume porosity ($V_p$) since $\sigma = (1 - V_p)^n$, where n was a shape factor which was 1.5 for sphere structure voids[27]. Previous work reported that physically compressing the MWCNT films would lead to a high $\sigma$ of ~1.08 MS/m[15]. However, it was hard to further improve the $\sigma$ with the compressing approach. It was known that CNT could form liquid crystals in CSA solutions at a proper concentration[28,29]. CSA treating approaches were often used to densify the CNT fibers to increase their $\sigma$[28,30], which, however, has not been reported for improving the $\sigma$ of CNT films yet. In the CNT fibers, CNTs were charged after being treated with CSA, which would possess many negative and positive sites. These charges sites increased the interaction between two CNTs through the Coulomb force. Therefore, the Coulomb force, together with the van der Waals force, led to highly packed CNTs in the fibers[31]. A developed CSA treatment method was created in this work.

Different from the previously reported dipping and drying at room temperature process, the HCl-MWCNT films were immersed in CSA and heated up to 150 °C in a culture dish until the added CSA was dry. The self-assembled structures formed in the CSA solution could be maintained in the MWCNT films after the CSA was dry. The detailed experimental procedure was described in the supplementary information. The obtained CSA-treated MWCNT films (CSA-MWCNT) exhibited a remarkable enhancement of $\sigma_\parallel$ up to $9.92 \pm 1.74$ MS/m. The $\sigma_\perp$ values of annealed-MWCNT, HCl-MWCNT, and CSA-MWCNT were also measured at the optimized conditions for their maximum $\sigma_\parallel$, which were 0.11, 0.57, and 3.89 MS/m (Fig. S8), respectively.

The thickness-dependent electrical conductivities of both pristine-MWCNT and CSA-MWCNT films in the parallel direction were shown in Fig. S9. MWCNT films with different thicknesses were obtained by changing the collection time. The thicknesses of the films were confirmed by the SEM images of the cross-section of MWCNT films (Fig. S10). Figure S9 showed that the $\sigma_\parallel$ of both pristine-MWCNT and CSA-MWCNT films increased with the decrease of the film thickness. The results were consistent with that reported in previous works for CNTs[32]. Figure S11 showed the dielectric functions of the CSA-MWCNT film. The Im($\varepsilon$) in the parallel direction was larger than that in the perpendicular direction. The result was similar to that reported in the previous work[33].

The maximum $\sigma_\perp$ was obtained in CSA-MWCNT, which was about two orders in magnitude higher than that of pristine-MWCNT. While it was only ~1/3 of the maximum $\sigma_\parallel$. The maximum $\sigma_\parallel$ of MWCNT films in this work was compared with that of CNT films in the literature (Fig. 1d). The electrical conductivity values were listed in Table S2. Generally, filtrated CNT films had a $\sigma$ of <0.5 MS/m. While winded CNT films had a $\sigma$ of <~1 MS/m. Recently a high $\sigma$ was reported to be ~1.08 MS/m for a compressed MWCNT film. The maximum $\sigma_\parallel$ was about ~10–20 times higher than that of CNT only and CNT composite films reported in the literature.

The dramatic enhancement in the $\sigma_\parallel$ and the $\sigma_\perp$ after CSA treatment was attributed to the densely packed structure of MWCNTs in the films, as demonstrated by the SEM images of the morphology and thickness of the CSA-MWCNT in Figs. S6, S7. The smooth and dense surfaces of CSA-MWCNT films demonstrated the dense packing of MWCNTs in the films. The reason for the densely packed of MWCNTs in the film was attributed to the zipping effect, as reported by Hata et al. that CNT forest would be packed densely since the zipping effect of liquid would draw tubes together[34]. For CSA-MWCNT films, the MWCNTs were drawn together by the van der Waals force during the evaporation of CSA. In addition, CSA was a strong acid with high hygroscopicity[8,35], which would polarize the MWCNTs[35] and remove the water molecules surrounding the CNT tubes[36] to increase the interaction between two MWCNTs. A transmission electron microscope (TEM) image of the CSA-MWCNT film was shown in Fig. 1e together with the inserted diffraction patterns, which were assigned to the (002) and the (100) according to the literature[34]. CSA-MWCNT contained ~11 wt% iron nanoparticles, which resided in the core of CNTs and on their side walls covered by crystalline graphite (Fig. S1). X-ray diffraction (XRD) results showed that a new peak appeared at $2\theta = 25.4°$, which corresponded to a d spacing of 0.31 nm, as shown in the inserted illustration image in Fig. 1f. The d spacing was close to the values of about 0.34 nm reported for self-assembled single-walled CNT in the literature[34,37], which was attributed to the (002)[34]. The densely packed MWCNTs in the films were illustrated in Fig. 1g. The number of CSA-MWCNT layers could be roughly obtained to be 217 (Fig. S12). The relative density of CSA-MWCNT films was calculated to be $91.65 \pm 0.66\%$, which indicated that MWCNTs were not perfectly packed in the films. A detailed calculation of the relative density was shown in the supplementary information.

The high $\sigma_\parallel$ of the MWCNT films makes them promising in potential applications such as electromagnetic interference (EMI)

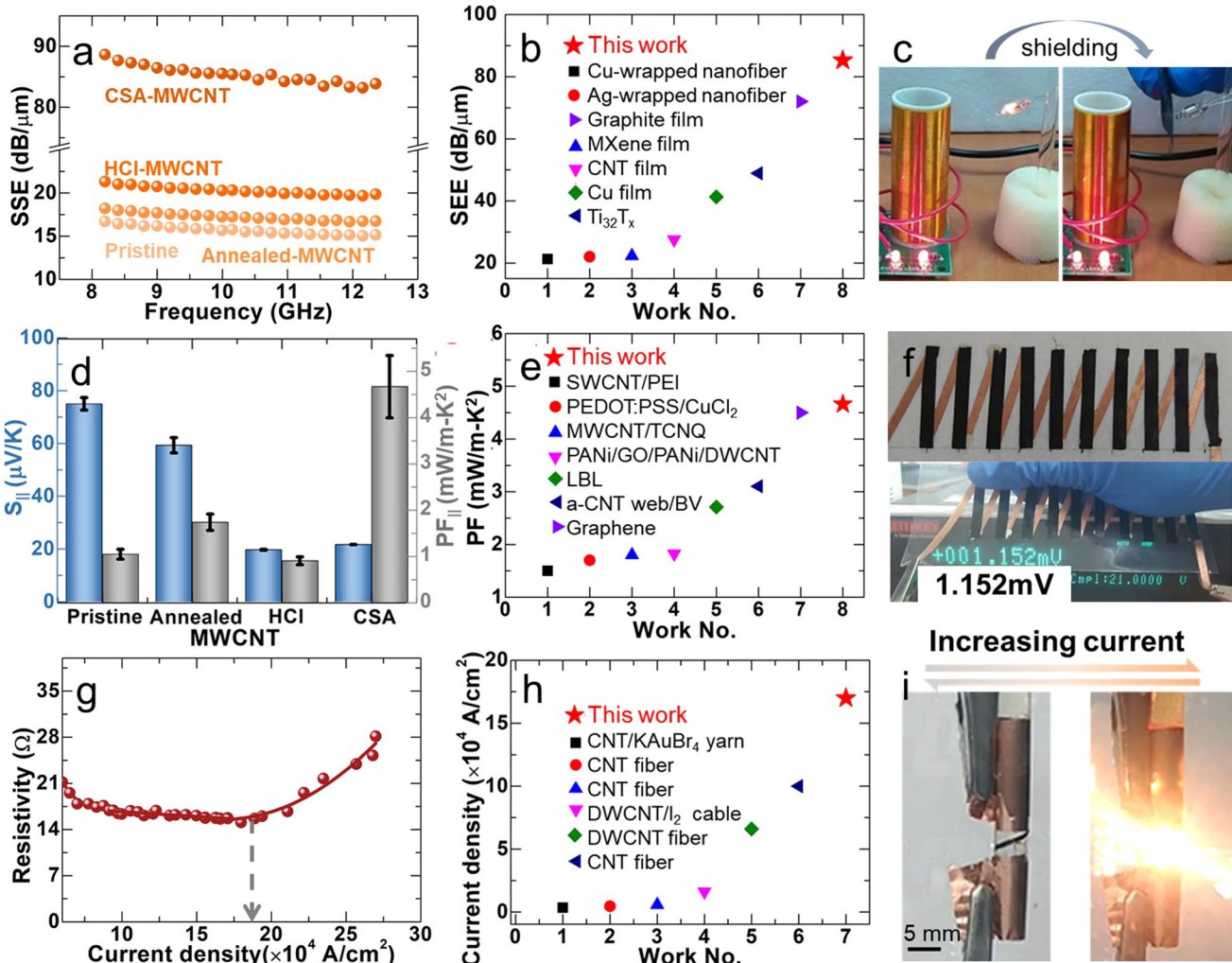

**Fig. 2 | Electromagnetic interference shielding, thermoelectric, and ampacity properties characterization of MWCNT films. a** SSE of the MWCNT films. **b** Comparison of the SSE(CSA-MWCNT) with that of state-of-the-art EMI shielding materials in the literature. **c** The optical images of a facial wireless power transmission device before and after shielding with CSA-MWCNT. **d** $S_{||}$ and $PF_{||}$ of the pristine-MWCNT, annealed-MWCNT, HCl-MWCNT, and CSA-MWCNT. **e** Comparison of the $PF_{||}$(CSA-MWCNT) with that of p-type MWCNT-based flexible

thermoelectric films in the literature. **f** The optical images of the TEG and the TE voltage generated by the human hand at a temperature difference of 5 K. **g** The resistivity of a piece of CSA-MWCNT film as a function of the current density. **h** Comparison of the current density of CSA-MWCNT films with that of CNT-only films/fibers in the literature. **i** The optical images of a piece of CSA-MWCNT as a filament before and after passing a current of 0.5 A.

shielding, thermoelectric (TE) techniques, high-ampacity conductor, etc. Figure 2a showed the specific shielding efficiency (SSE = shielding efficiency/thickness) of the MWCNT films. The SSE values of the MWCNT films were in the order of SSE(CSA-MWCNT) > SSE(HCl-MWCNT) > SSE(annealed-MWCNT) > SSE(pristine-MWCNT) in the frequency range of 8–12 GHz since SSE values were proportional to the electrical conductivity of a material according to Simon's equation[38]: $SSE = (50 + 10\log\left(\frac{\sigma}{f}\right) + 1.7d(\sigma f))/d$, where f and d were the frequency and the thickness of the sample, respectively. The maximum SSE value was SSE(CSA-MWCNTs) = ~85 dB/μm since CSA -MWCNT films had a higher electrical conductivity than other MWCNT films. The SSE(CSA-MWCNTs) was compared with state-of-the-art EMI shielding materials in the literature, as shown in Fig. 2b. It showed that SSE(CSA-MWCNTs) was about 1.2–4.0 times higher than that of the previously reported state-of-the-art graphene films, the MXene films, and even metal films, etc.[8,39–41]. The related values were listed in Table S3. Figure 2c demonstrated that a piece of CSA-MWCNT film successfully shielded the electromagnetic waves generated by a Tesla coil of 50 Hz. The EMI shielding was also recorded in Supplementary Video 1. The results indicated that the highly conductive CSA-MWCNT film would be promising in developing high-performance EMI interface materials.

TE properties were tested to show the potential of these MWCNT films in thermoelectric applications. Figure 2d showed the Seebeck coefficients of the pristine-MWCNT, annealed-MWCNT, HCl-MWCNT, and CSA-MWCNT in the direction parallel to the winding direction, which were in the order of $S_{||}$(pristine-MWCNT) > $S_{||}$(annealed-MWCNT) > $S_{||}$(HCl-MWCNT) > $S_{||}$(CSA-MWCNT). The positive values indicated that the obtained MWCNT films were all p-type materials. After the annealing and HCl treatment, the $S_{||}$ decreased while the $\sigma_{||}$ increased. There was a trade-off relationship between $S_{||}$ and $\sigma_{||}$ for TE materials. However, it was noticed that the $S_{||}$ maintained at about 20 μV/K in the following densification process, although the $\sigma_{||}$ kept increasing. Similar results were observed in previously reported TE films, which were densified by compressing[15]. Because more CNT would lay down to increase the mean distance according to Mott's theory, thus increasing the electrical conductivity, which would not affect the S of MWCNT since S was isotropic for the one-dimensional CNTs[42]. The increased $\sigma_{||}$ and maintained $S_{||}$ led to a high $PF_{||}$ of 4.66 mW/m K$^2$ in CSA-MWCNT. This $PF_{||}$ value was superior to that of non-inorganic materials involved in films in the literature (Fig. 2e). Related values were listed in Table S4. The average thermal conductivities for pristine-MWCNT and CSA-MWCNT were 21.93±0.10 and

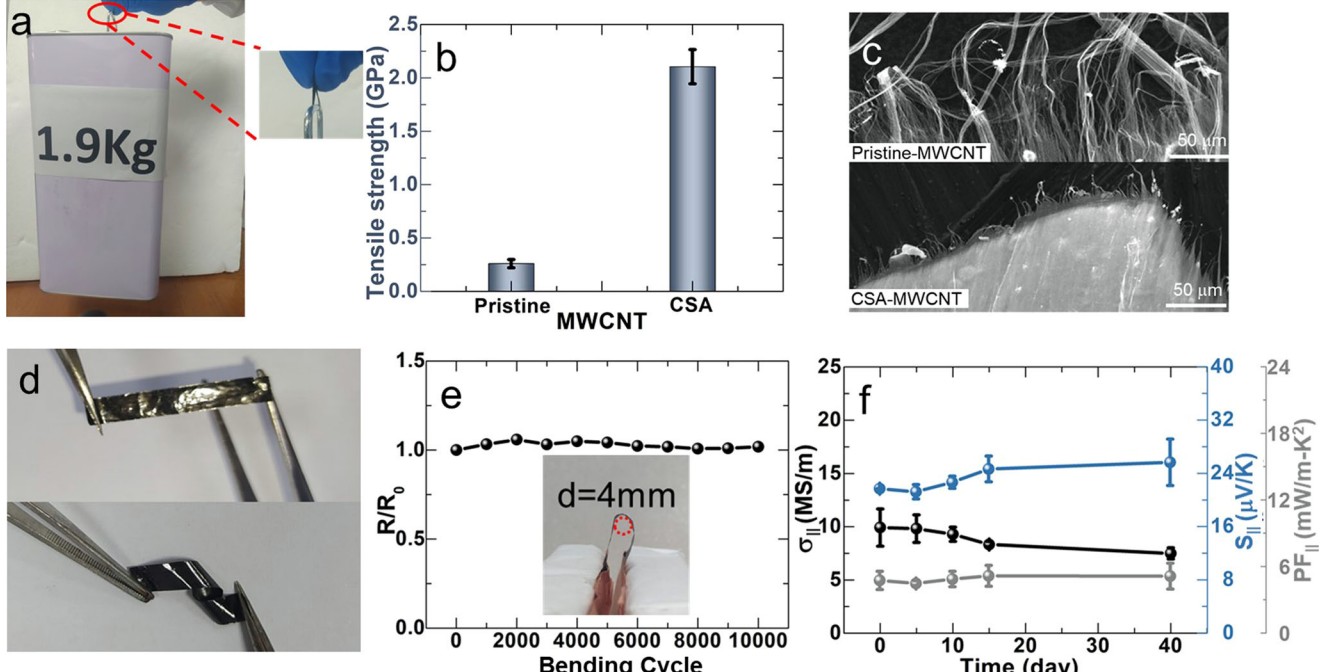

**Fig. 3 | Mechanical properties and stability characterization of MWCNT films.** **a** The optical images of a water bottle of ~1.9 Kg lifted by CSA-MWCNT film. The tensile strength (**b**) and SEM images of fracture surfaces (**c**) of pristine-MWCNT film and CSA-MWCNT film. **d** The optical images of twisted CSA-MWCNT stripes. **e** The $\sigma_{\parallel}$ of a CSA-MWCNT stripe as a function of the bending cycle. **f** The $PF_{\parallel}$, $\sigma_{\parallel}$ and $S_{\parallel}$ of CSA-MWCNT film in the air as a function of storing time.

45.90 ± 0.10 W/m-K, respectively (Table S5). It was interesting to note that the figure-of-merit of CSA-MWCNT films has increased by three times as compared to that of pristine-MWCNT films. The obtained materials would be promising for active cooling (Peltier effect) as suggested by Kono et. al. in the literature, where high power factor and high thermal conductivity were required[43,44]. $PF_{\perp}$ values were also measured for the MWCNT films, which were in the order of $PF_{\perp}$ (CSA-MWCNT) > $PF_{\perp}$ (HCl-MWCNT) > $PF_{\perp}$ (annealed-MWCNT) > $PF_{\perp}$ (pristine-MWCNT) (Fig. S13). The maximum $PF_{\perp}$ value was only half of the $PF_{\parallel}$(CSA-MWCNT). A thermoelectric generator (TEG) was fabricated with ten pairs of legs to demonstrate the heat-to-electricity conversion ability of the obtained high $PF_{\parallel}$ CSA-MWCNT films (Fig. 2f). It could generate an output voltage of 1.2 mV when being held by human hands at a temperature difference of ~5 K (Fig. 2f). The TEG could generate a short-circuit current and an open-circuit voltage were ~0.4 mA and ~13.1 mV, respectively, at a temperature difference of 60 K. The maximum output power was ~1.2 μW when the loading resistance equaled to the inner resistance of 35 Ω for the fabricated TEG at the temperature difference of 60 K. Detailed characterizations of the TEG was shown in Fig. S14.

The ampacity of the high $\sigma_{\parallel}$ CSA-MWCNT films was tested to evaluate their potential application of them as high-ampacity conductors. Figure 2g showed the resistivity of a piece of CSA-MWCNT film as a function of the current. In the beginning, the resistivity decreased with the current, which should be due to the reorganization of the embedded iron particles, according to the literature[45]. Then the resistivity remained unchanged up to $1.7 \times 10^5$ A/cm$^2$, after which it increased exponentially. The ampacity was defined as the maximum current density where the resistivity remained unchanged. The ampacity of CSA-MWCNT film is about 1.7 times higher than the record for CNT-only films/fibers in the literature (Fig. 2h), which approached the highest value reported for individual CNTs ($10^9$ A/cm$^2$)[46–48] and CNT-metal composite ($10^7$ A/cm$^2$)[45]. The lower ampacity of CSA-MWCNT film should be attributed to the higher electrical resistivity and lower thermal conductivity derived from the non-uniformity of the film compared to single tubes and SWCNT-Cu composites. Because

CSA-MWCNT films contained irregular iron nanoparticles that led to lower electrical conductivity and thermal conductivity. The higher electrical and thermal conductivity in single tubes and SWCNT-Cu reduced the material temperature by generating less Joule heating and by dissipating the heat efficiently, preventing the breaking of the sample[28]. Related values were listed in Table S6. Figure 2i showed that a piece of CSA-MWCNT film with a length of ~30 mm and a width of ~1 mm could work as a filament. When increasing the current to 0.5 A, white light could be produced. The process was reversible, as shown in Supplementary Video 2, which indicated the high ampacity and the good thermal stability of the CSA-MWCNT film.

The mechanical properties of the CSA-MWCNT films were studied. Figure 3a showed that a piece of CSA-MWCNT film with a width of 9 mm and thickness of ~1 μm could pull up a water bottle of ~1.9 kg. More details could be seen in Supplementary Video 3. The strain-stress curves of pristine-MWCNT and CSA-MWCNT films were shown in Fig. S15. The tensile strength of the CSA-MWCNT film along the winding direction was measured to be ~2 GPa, which was about eight times higher than that of pristine-MWCNT films (Fig. 3b). The tensile strength of CSA-MWCNT in the direction perpendicular to the winding direction also increased by about 16 times, which increased from 0.014 to 0.218 GPa (Fig. S16). The increased tensile strength in both the parallel and the perpendicular direction to the winding direction demonstrated the increase of interactions between two carbon nanotubes in the CSA-MWCNT Fig. 3c showed the SEM images of the breaks of pristine-MWCNT films and CSA-MWCNT films. Although some long MWCNTs pulled out from the pristine-MWCNT film could be observed, the spare structure and weak inter-tube frication resulted in both low tensile strength and elongation at the break. The low strength of the pristine-MWCNT film was ascribed to the loose stacking and weak interaction between tubes, as shown in Fig. 3c and S6. Unlike the pristine-MWCNT film with a relatively loose surface, the CSA-MWCNT film showed compact edges, helping arrest crack propagation. The individual MWCNTs or MWCNT bundles were pulled out from the CSA-MWCNT film with strong tube-tube friction, and it increased the mechanical strength.

Besides, the CSA-MWCNT films also exhibited good flexibility. Figure 3d shows that a piece of the CSA-MWCNT film can be twisted freely. The CSA-MWCNT film was a shiny metallic luster which was different from the dark-black pristine-MWCNT film. The $\sigma_{\parallel}$ as a function of the bending cycle was recorded as shown in Fig. 3e. No obvious decay and fluctuations were observed in the resistance when a piece of CSA-MWCNT film was bent 180° at a bending curvature diameter of 4 mm for 10,000 cycles. In addition, the $PF_{\parallel}$ of the CSA-MWCNT films could maintain over 40 days when they were stored in the air (Fig. 3f) with a slight decrease in $\sigma_{\parallel}$ and an increase in $S_{\parallel}$. The results demonstrated that the CSA-MWCNT films possessed high mechanical properties and their PF even increased by 8% for a long time of 40 days due to the increase of the Seebeck coefficient of CSA-MWCNT film caused by the de-doping effects in the air.

A radar plot was prepared to compare the properties of CSA-MWCNT film with state-of-the-art materials such as silver films, $Bi_2Te_3$ films, carbon fibers, etc. (Fig. S17). The related data were listed in Tables S4, S7, S8. CSA-MWCNT film possesses high specific electrical conductivity of $5.26 \times 10^4$ Scm$^2$/g comparable to that of silver films $(5.72 \times 10^4$ Scm$^2$/g)[49] and much higher specific shielding efficiency of 85.5 dB/μm than that of silver films (5.849 dB/μm)[49]. It also possesses a high power factor (4660 μW/m·K$^2$) close to that of state-of-the-art $Bi_2Te_3$ films (4700 μW/m·K$^2$)[50] and good mechanical properties (1.15 N/tex) approaching that of carbon fibers (4.08 N/tex)[51]. The results indicated that CSA-MWCNT film was a promising multifunctional material, which may be used in electromagnetic interference shielding and flexible thermoelectric, etc.

In summary, winded MWCNT films were synthesized, which exhibited ultra-high electrical conductivity of $9.92 \pm 1.74$ MS/m in the direction parallel to the winding direction after being annealed, purified, and densified. The high $\sigma_{\parallel}$ was about ~10–20 times higher than that of CNT films reported in the literature, which was even comparable to that of metals. The main reason for the high electrical conductivity was attributed to the highly densified packing structure of MWCNT in the film derived from CSA enhanced zipping effect. As demonstrated by the new peak in the XRD spectrum and the diffraction patterns in the TEM images, MWCNTs formed crystalline structures in the CSA-MWCNT film. The high $\sigma_{\parallel}$ resulted in a high SSE(CSA-MWCNT) of ~1.2–4.0 times higher than that of state-of-the-art EMI shielding materials, a high $PF_{\parallel}$ value superior to that of MWCNT-based thermoelectric materials films, and high ampacity of over 1.7 times higher than that of CNT-only films/fibers reported in the literature. In addition, the CSA-MWCNT film also exhibited a high strength of ~2 GPa in the direction parallel to the winding direction. The results demonstrated the achievements of a combination of lightweight, high strength, and high electrical conductivity in macroscopic multifunctional CNT films with a potentially scalable manufacturing process.

## Methods
### Materials
Ethanol was purchased from Tianjin Fuyu Fine Chemical Co., Ltd., China. Methanol was purchased from Sinopham Chemical Reagent Co., Ltd., China. N-hexane was purchased from Tianjin Zhiyuan Chemical Co., Ltd., China. Thiophene and ferrocene were Meryer Chemical Technology Co., Ltd., China. Hydrochloric acid(HCl) was purchased from Kelong Chemical Co., Ltd., China. Sulfurochloridic acid (CSA) was purchased from Meryer Chemical Technology Co., Ltd., China. All the chemicals were used as received.

### Synthesis of aerogel
The aerogel was continuously synthesized at 1300–1500 °C in a horizontal furnace by using an alumina tube with a diameter of 60 mm as the reactor. Nitrogen was used as the carrier gas. The precursor solution was made by dissolving ferrocene and thiophene in a mixture of n-hexane and ethanol. This solution was injected into the reactor at a rate of 1–2 mL/min, which was then carried into the high-temperature zone by $N_2$ at a flow rate of 0.5–1 L/min. The aerogel was driven out from the reactor to the air atmosphere by the enclosed $N_2$ and collected by a roller.

### Formation of CNT Film
A winding drum with a diameter of 10 cm was used for the CNT collection. The winding drum was wrapped with the paper substrate. When the CNT aerogel was collected on the winding drum, ethanol was sprayed on it. The amount of the solution was controlled by the spraying time. The winding rate was tuned at a speed of 3 mm/s in the preparation process. When the collection was complete after 3, 5, 7, 9, and 11 h, respectively, the film was peeled off from the drum for different thicknesses. Then it was compressed for 5 min with a pressing machine (Kejing MSK-2150 China) at a force of 4 tons to make the surface of the MWCNT films smooth. After that, the obtained pristine-MWCNT films were dried in a vacuum at a temperature of 50 °C and kept in the glove box before use.

### Treatment of CNT film
The obtained pristine-MWCNT was heated in a tube furnace under an inert atmosphere for a certain period (annealed-MWCNT), and it was then treated with concentrated HCl (HCl-MWCNT). The HCl-MWCNT was soaked in deionized water for about 5 min, then dried at room temperature. After rinsing with deionized water and drying in the air, the film was placed flat on the bottom of the petri dish and treated with CSA for densification at 150 °C.

### Characterization
The sizes of MWCNT films used for property tests were summarized in Table S9. The measurement of the Seebeck coefficient and electrical conductivity was performed with commercial equipment (NETZSCH SBA-458, Germany). The electrical conductivities were obtained with a four-probe method, as shown in Fig. S18. A detailed description of the electrical conductivity measurement was shown in the supplementary information. The Seebeck coefficients were obtained simultaneously with the electrical conductivities by a two-way heating test approach, the values of which were determined by taking the average of ≥100 data points. All the tests were performed under Ar protection at room temperature and the samples were cut into strips with a length of 20 mm and a width of 5 mm.

TGA of the MWCNT films was measured under the air atmosphere by Synchronous Thermal Analyzer Q600 at a heating rate of 10 °C min$^{-1}$ Scanning electron microscope (SEM) images were obtained using FEI QUANTA 250 FEG, USA. Transmission electron microscope (TEM) images were obtained with a JEOL 2010D, Japan, accelerating voltage, 200 kV. Thermogravimetric analyses (TEG) of MWCNT films were performed under the air atmosphere using a synchronous Thermal Analyzer Q600, USA, at a heating rate of 10 °C/min. XRD data were performed with Bruker D8 ADVANCE, Germany. X-ray photoelectron spectroscopy was performed with Thermo Fisher ESCALAB Xi +, USA. Raman spectra were recorded with a Thermo fisher Raman spectrometer, USA, with an excitation wavelength of 514 nm. as shown in the supplementary information (Fig. S3). The EMI shielding performance was measured by Network Analyzer (Agilent PNA 5227 A). The output voltage of was obtained using a Keithley 2400 Multimeter, USA. The CNT samples were cut into strips with an aspect ratio of 2 (a length of 10 mm and a width of 5 mm). And their mechanical properties were tested by DMA850 (TA Instruments, USA) with a crosshead speed of 0.01 N/min. The thermal conductivities of pristine-MWCNT and CSA-MWCNT in the in-plane direction were measured by a laser flash method with NETZSCH LFA-467 (German). The dielectric function of the film was measured by an ellipsometer (HORIBA UVISEL PLUS, Japan).

## Data availability

All other data that support the plots within this paper and other findings of this study are available from the corresponding authors upon reasonable request. Source data are provided with this paper.

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

## Acknowledgements

H.W. acknowledges financial support from the National Natural Science Foundation of China (grant numbers: 52276014, 51876151, and 51888103), the Key Research and Development Program in Shaanxi Province of China (grant number: 2021GXLH-Z-056), the Fundamental Research Funds for the Central Universities, the World-Class Universities (Disciplines), and the Characteristic Development Guidance Funds for the Central Universities (grant number: PY3A010) and the start-up funding from Xi'an Jiaotong University (grant number: QY1J003). This work is also supported by the HPC platform and the Instrument Analysis Center, at Xi'an Jiaotong University. We thank Miss Dan He at the Instrument Analysis Center of Xi'an Jiaotong University for their assistance with the Raman spectrum analysis.

## Author contributions

H.W. and X.S. contributed to the manuscript equally, conceived and planned the experiments; X.S. carried out the experiments; Y.W., K.L., J.W., and X.D. contributed to sample preparation; H.W. performed the analysis and wrote the manuscript; B.C., D.C., L.Z., and J.Y. contributed to the interpretation of the results; All authors discussed the results and commented on the manuscript.

## Competing interests

The authors declare no competing interests.
