## [Peer Review File · Nature Communications]

REVIEWER COMMENTS

Reviewer #1 (Remarks to the Author):

The manuscript entitled “Acid enhanced zipping effect to densify MWCNT packing for strong, light, multifunctional MWCNT films with ultra-high electrical conductivity” details a method to produce aligned CNT films and their resultant properties. The manuscript claims the highest reported electrical conductance for MWCNT films (9.92 MS/m). This is perhaps the most significant claim of the manuscript, and it would be of great importance to applications related to multi-functional and light-weight composite materials and energy applications. The manuscript demonstrated many potential applications of the films which combined strength, flexibility, conductivity, and thermoelectric properties. In my opinion, the manuscript lacked significant details that make repeating and fully understanding the results not possible. A list of concerns is listed below. Because of these concerns, I recommend significant revisions before further consideration.

1. The physical characterization of the CNTs is lacking. A CNT diameter distribution is found in the supporting information, but not mentioned in the main document. What is the length distribution of the CNTs? How does the length distribution compare to the electron mean free path? How does the CNT length compare to the length scale at which conductivity measurements were obtained?
2. The TGA curves found in the supporting information suggest that iron comprises 37 % of the as-synthesized CNT product by mass. This is, perhaps, not surprising because of the floating catalyst synthesis method. What is the residual iron mass after the acid treatment and annealing? This is quite important, as it provides context relative to the role of iron in the observed conductivity increase.
3. The TEM images in Figure 1e suggests that a relatively large amount of iron remains. Does the iron reside in the core of the CNTs or on their side walls?
4. The parameters and equations used to calculate the electrical conductivity are not provided. How far apart were the measurement electrodes? What was the cross-sectional area of the sample, and how uniform was the area over the length of the device? Was the voided area of the CNT cylinder included in the volume calculation? If the voided area is not considered in the calculation, then the reported results might be quite misleading.

5. How were the electrical connections to the CNT sample established for conductivity measurements? What was the composition of the electrodes?

6. Similar concerns to 3 are raised with respect to mechanical measurements. What was the length of the sample tested? What was the aspect ratio of the sample? Could an estimated strength calculation be made by assuming a close packing of CNTs pulled axially and opposed by van der Waals force? This would lend credibility to the claim that the strength was a result of this mechanism (line 267).

7. How dense were the films after crystallization? This measurement should be readily achievable and will help the reader determine the packing density of the CNT film. While XRD data suggests close packing, why not include a readily obtained direct measurement?

8. If the films are crystalline and aligned, approximately how many stacked CNT layers existed in the tested samples?

9. The acronym "CSA" was never defined.

10. In line 132, why is a "~" symbol present when reporting conductivity, followed by 3 significant digits?

11. Line 222: a temperature gradient has units of K/m.

12. Was the Raman spectroscopy laser polarized in a specific orientation relative to the fiber orientation (Figure S2)? If yes, what was the polarization? Of no, how may the differences in Raman spectra with respect to orientation be justified?

13. Numerous grammatical errors are present in the manuscript. I would recommend a thorough review from a native English speaker.

Reviewer #2 (Remarks to the Author):

The authors studied the electrical conductivity of multifunctional carbon nanotube (CNT) films synthesized by a floating chemical vapor deposition method. The synthesized CNT films were purified at high temperatures and treated with concentrated hydrochloric (HCl) acid. The authors report that the densified structure of carbon nanotubes shows high electromagnetic interference shielding efficiency, high thermoelectric power factor, and high ampacity. The authors used ChloroSulfonic Acid (CSA) treatment method. The authors also immersed the HCl-Multi Wall CNT films in CSA and heated them up to 150⁰C in a culture dish until the added CSA was dry. This approach is a simple modification of a previously reported dipping and drying at room temperature process. The research work is nice. However, I do not see this approach as innovative, and the findings are not that promising that it can be published in nature communications. In addition to that, I have some questions for the authors.

1. Properties of self-assembled carbon nanotubes (works is for Single Wall CNT, not for Multi-Wall CNT, however) are presented in Nano Lett. 19, 5, 3131–3137 (2019) by Roberts et al. analyzing dielectric functions. The dielectric function and conductivity are related to each other through a well-known equation. Smaller ϵ implies larger σ . How much do your findings agree?
2. In the literature, it is also noted that the dielectric response and the electrical conductivity of CNT films are highly anisotropic, and low conductivity in a particular direction is due to the depolarization effect [for example, Bondarev et al. Phys. Rev. Applied 15, 034001 (2021)]. The authors are missing a core concept.
3. The authors have used the term MWCNT in many places in the manuscript. However, in several places, the authors only write CNT. Which CNT authors are talking about in these places? Are these CNT Single Wall CNT or Multi-Wall CNT?
4. The authors mentioned that the winding drum to collect CNT was a diameter of 10 cm. What about the size of CNT they used and the MWCNT they prepared?
5. It is said that collecting CNT in winding drum pristine-MWCNT was prepared. I am bothered with the word pristine as there is no clear figure of the sample showing MWCNT, and the authors have very little discussion about this.
6. The literature is rich in the study of thickness-dependent properties of CNT film. How about the thickness of your film? How the conductivity of your film depends on film thickness?

Reviewer #3 (Remarks to the Author):

The authors report on a dry-drawn, and chemically-treated MWCNT sheet showing multi-functionalities including high EMI shielding, thermoelectric power factor, current ampacity, and mechanical strength. This reviewer considers a part of properties demonstrated here (e.g. ultrahigh electrical conductivity) is interesting as a carbon-based nanomaterial while its comparison with standard materials such as silver films for EMI, Bi₂Te₃ for thermoelectrics, and carbon fibers/CFRP for mechanics is fair. Several additional points are noted below.

1. The authors claimed "light". They should note the quantitative density (g/cc) of each material.
2. Thermal conductivity, along with power factor, is important for showing thermoelectric efficiency.
3. This reviewer did not understand the mechanism of CSA-driven densification. The authors should explain it around ~line 122 after the introduction of reference 25.
4. Long-term stability data (Figure 3f) show the gradual de-doping, where electrical conductivity decreased and the Seebeck coefficient slightly increase. Additionally, CAS seems to show moderate volatility, potentially leading to gradual de-doping. As such, this reviewer did not approve for the statement of line 286-288 "The results ... with stable electrical and thermoelectric properties.
5. Mechanical properties for perpendicular direction against alignment should be noted.
6. The ampacity presented here was much lower than single tubes and SWCNT-Cu composites, as noted in the manuscript. Could you describe its reason in the manuscript?

Several mistypes should be checked. For example,

Line 205: the Wiedemann-Franz law => (other physics law?)

* The W-F law explains the relationship between the electrical and thermal conductivity of ideal metals. This is not associated with the Seebeck coefficient.

Line 222: generat => generate

Rebuttal

Manuscript ID: NCOMMS-22-30697-T

Title: Acid enhanced zipping effect to densify MWCNT packing for strong, light, multifunctional MWCNT films with ultra-high electrical conductivity

Reviewer Comments to Author:

Referee 1

General comments:

The manuscript entitled “Acid enhanced zipping effect to densify MWCNT packing for strong, light, multifunctional MWCNT films with ultra-high electrical conductivity” details a method to produce aligned CNT films and their resultant properties. The manuscript claims the highest reported electrical conductance for MWCNT films (9.92 MS/m). This is perhaps the most significant claim of the manuscript, and it would be of great importance to applications related to multi-functional and lightweight composite materials and energy applications. The manuscript demonstrated many potential applications of the films which combined strength, flexibility, conductivity, and thermoelectric properties. In my opinion, the manuscript lacked significant details that make repeating and fully understanding the results not possible. A list of concerns is listed below. Because of these concerns, I recommend significant revisions before further consideration.

Author’s response:

Thank you very much for your time and efforts. We sincerely appreciate it.

Single carbon nanotubes (CNTs) have excellent electrical and mechanical properties, which have attracted much attention since it was reported in 1991. However, the electrical and mechanical properties of CNT films are far away behind those of single CNTs because of the difficult material process. In this work, we reported a scalable method for multiwall carbon nanotube (MWCNT) film preparation. The obtained MWCNT films exhibited a high electrical conductivity of 9.92 ± 1.75 MS/m and high strength of 2.10 ± 0.15 GPa, which led to the high performance of these MWCNT films in practical applications such as electromagnetic interference shielding, thermoelectrics, *etc.* A detailed description has been added point-to-point in the revised manuscript according to the reviewer’s comments. It’s believed that the readers may repeat and understand the results without problems. We sincerely appreciate the reviewer’s help and we would be very glad to provide more details upon request.

Reviewer's Comments 1:

The physical characterization of the CNTs is lacking. A CNT diameter distribution is found in the supporting information, but not mentioned in the main document. What is the length distribution of the CNTs? How does the length distribution compare to the electron mean free path? How does the CNT length compare to the length scale at which conductivity measurements were obtained?

Author's response:

Thank you very much for your good suggestions.

Additional experiments have been performed to evaluate the length of the MWCNTs with transmission electron microscopy. And the description of the physical characteristics and properties has been added in the revised manuscript.

It is very challenging to obtain the accurate length of CNTs synthesized with a floating catalytic chemical vapor deposition method as suggested by Zhou *et. al.*, (**Mater. Des.**, 2021, 203, 109557). Because the CNTs or CNT bundles growing from the same catalyst particle will be aggregated to form very complex three-dimensional networks during the growth of CNTs with the floating catalytic chemical vapor deposition method. The length of the MWCNTs/MWCNT bundles was roughly estimated to be about 20 μm as shown in the transmission electron microscope images in **Figure S2**, which is similar to the values ($<20 \mu\text{m}$) reported previously (**Nano Lett.**, 2016, 16, 946; **Adv. Funct. Mater.**, 2022, 32, 2103397). The length of the MWCNTs/MWCNT bundles in this work is much larger than the mean free path of 0.27-0.55 μm for MWCNTs reported by Chai *et. al.* (**Nanotechnology**, 2010, 21, 235705).

A four-probe method was used for the electrical conductivity measurement with strip samples at a length of 20 mm and a width of 5 mm, which is similar to the previous work (**ACS Energy Lett.**, 2021, 6, 4355–4364). The distance between each electrode is 3-8.5 mm. Therefore, the length of the MWCNTs/MWCNT bundles in this work is much smaller than the length scale at which the electrical conductivity measurements were obtained.

Related content has been added to the revised manuscript.

Figure S2 TEM images of MWCNTs/MWCNT bundles for the length evaluation

Reviewer's Comments 2:

The TGA curves found in the supporting information suggest that iron comprises 37 % of the as-synthesized CNT product by mass. This is, perhaps, not surprising because of the floating catalyst synthesis method. What is the residual iron mass after the acid treatment and annealing? This is quite important, as it provides context relative to the role of iron in the observed conductivity increase.

Author's response:

Additional experiments have been performed to get the content of the residual iron after the acid treatment and annealing.

All the TGA experiments were performed in the air. In **Figure S4**, assuming all the iron was oxidized to be Fe_2O_3 , the content of iron by mass for the as-synthesized MWCNT film was calculated to be 28.7 wt.%. After annealing, the annealed MWCNT still contained almost the same amount of Fe_2O_3 as shown in **Figure S4**, indicating that the amorphous carbon was negligible. After HCl treatment, the content of iron decreased to ~11 wt.% because iron without being covered fully by crystalline graphite or MWCNTs has reacted with HCl. The content of iron maintains after further chlorosulfonic acid treatment, indicating the residual iron was all covered by crystalline graphite or embedded into MWCNTs.

We agree with the reviewer that the residual iron will contribute to the increase of the electrical conductivity of MWCNT films. Similar results have been observed in the literature. The electrical conductivity of CNTs was enhanced to 0.04-1.02 MS/m by introducing Au/Pt nanoparticles (Carbon, 2019, 141, 497-505) and iron nanoparticles

inside of CNTs (*Adv. Energy Mater.*, 2022, 12, 2200256; *Adv. Funct. Mater.*, 2022, 32, 2203080). However, the electrical conductivity of previous CNT films is lower than that of MWCNT films in this work. The lower electrical conductivity in previous works indicates that the ultra-high electrical conductivity of the MWCNT film in this work is not mainly due to the residual iron in the MWCNT film (*Adv. Funct. Mater.*, 2022, 32, 2203080).

Related content has been added to the revised manuscript.

Figure S4 TGA spectra of the pristine-MWCNT, annealed-MWCNT, HCl-MWCNT and CSA-MWCNT.

Reviewer's Comments 3:

The TEM images in Figure 1e suggests that a relatively large amount of iron remains. Does the iron reside in the core of the CNTs or on their side walls?

Author's response:

These iron nanoparticles reside both in the core and on the side walls of the CNTs as demonstrated by the TEM images in **Figure S1**.

Related content has been added to the revised manuscript.

Reviewer's Comments 4:

The parameters and equations used to calculate the electrical conductivity are not provided. How far about were the measurement electrodes? What was the cross-sectional area of the sample, and how uniform was the area over the length of the device? Was the voided area of the CNT cylinder included in the volume calculation? If the voided area is not considered in the calculation, then the reported results might be quite misleading.

Author's response:

We truly appreciate the reviewer's time and efforts.

The electrical conductivity of film was measured by commercial equipment

(NETZSCH SBA-458, Germany) with a four-probe method. All the measurements were performed under Ar protection at room temperature and the MWCNT film samples were cut into strips with a length of 20 mm and a width of 5 mm. The distance between the electrodes can be seen in the following **Figure S14**, which is about 3-8.5 mm. The electrical conductivity (σ) of the sample was calculated with the following equation: $\sigma=1/\rho= L/(R \cdot A)$, where ρ is the resistivity, L is the length of the sample between electrode 2 and electrode 3, R is the resistance of the sample between electrode 2 and electrode 3, A is the cross-sectional area of the sample.

The cross-sectional areas were $0.0417 \pm 0.0046 \text{ mm}^2$, $0.0353 \pm 0.0017 \text{ mm}^2$, $0.0332 \pm 0.0010 \text{ mm}^2$, and $0.0032 \pm 0.0002 \text{ mm}^2$ for pristine-MWCNT films, annealed-MWCNT films, HCl-MWCNT films and CSA-MWCNT films, respectively. The cross-section area value was calculated with the equation: $A = \text{width of the sample} \times \text{thickness of the sample}$. This cross-sectional area was pretty uniform with relatively small error bars as shown in the following **Table S1** since the pristine-MWCNT films were compressed before used. These films had a relatively flat surface as shown in **Figure S6**. The voided area of the CNT cylinder was included in the cross-sectional area calculation.

Figure S14 (a) optical image of the sample holder for the electrical conductivity measurement; (b) Illustration of the electrical circuit for the electrical conductivity measurement with a four-probe method.

Table S1 The electrical conductivities and thicknesses of CNT films

Sample	Thickness (μm)	Electrical conductivity σ_{\parallel} (MS/m)	Electrical conductivity σ_{\perp} (MS/m)
Pristine-MWCNT	8.34 ± 0.92	0.18 ± 0.03	0.04 ± 0.006
Annealed-MWCNT	7.07 ± 0.35	0.49 ± 0.023	0.11 ± 0.006
HCl-MWCNT	6.64 ± 0.19	2.33 ± 0.24	0.56 ± 0.07
CSA-MWCNT	0.64 ± 0.05	9.92 ± 1.75	3.88 ± 0.42

Related content has been added to the revised manuscript.

Reviewer's Comments 5:

How were the electrical connections to the CNT sample established for conductivity measurements? What was the composition of the electrodes?

Author's response:

We truly appreciate your time and efforts.

The electrical conductivity was measured by commercial equipment (NETZSCH SBA-458, Germany) with a four-probe method. The electrodes are made of rhodium which contacted directly with the samples. The pressure was applied to ensure good contact between the sample and the rhodium electrodes by a pressure disk and the knurled nuts as shown in **Figure S14**. The electrical contact resistance between the electrodes and the samples was eliminated by the four-probe method.

Figure S14 (a) optical image of the sample holder for the electrical conductivity measurement; (b) Illustration of the electrical circuit for the electrical conductivity measurement with a four-probe method.

Related content has been added to the revised manuscript.

Reviewer's Comments 6:

Similar concerns to 3 are raised with respect to mechanical measurements. What was the length of the sample tested? What was the aspect ratio of the sample? Could an estimated strength calculation be made by assuming a close packing of CNTs pulled axially and opposed by van der Waals force? This would lend credibility to the claim that the strength was a result of this mechanism (line 267).

Author's response:

The CNT samples were cut into strips with an aspect ratio of 2 (a length of 10 mm and a width of 5 mm). And their mechanical properties were tested by DMA850 (TA Instruments, USA) with a crosshead speed of 0.01 N/ min. We are sorry that we are not capable of performing the theoretical calculation of the strength currently, which is out of our expertise.

The mechanism for the high strength of the MWCNT film was proposed according

to literature (**ACS Nano**, 2020, 14, 14134–14145). The high strength of the CNT films was ascribed to the excellent mechanical strength of individual CNTs and the strong interactions between them. After being treated with CSA acid, individual CNTs were charged, which led to many negative and positive sites on the CNTs due to the charge separation. These charged sites increased the interaction between two CNTs through Coulomb forces. In the meanwhile, the van der Waals force between two CNTs also played an important role in the interaction between two CNTs. The Coulomb force together with the van der Waals force thus led to the high strength of the MWCNT film. A similar mechanism has been proposed in the literature (**ACS Nano**, 2020, 14, 14134–14145).

Related content has been added to the revised manuscript.

Reviewer's Comments 7:

How dense were the films after crystallization? This measurement should be readily achievable and will help the reader determine the packing density of the CNT film. While XRD data suggests close packing, why not include a readily obtained direct measurement?

Author's response:

Thank you very much for your good suggestion.

The MWCNT films contain iron nanoparticles. The theoretical density of the MWCNT film is in the range of 2.08-2.11 g/cm³ (Assuming that all the iron particles are inside the carbon nanotubes, the theoretical density of MWCNT film containing 28.7 wt.% iron is 2.11 g/cm³; Assuming that all the iron particles are on the sidewalls of the carbon nanotubes, the theoretical density of MWCNT film containing 28.7 wt.% iron is 2.08 g/cm³).

The practical density of MWCNT films can be calculated with the equation: density = mass/volume. The mass of the sample was obtained with a high-precision microbalance and the volume of the sample was estimated by the following equation: volume = length x width x thickness. The practical density of CSA-MWCNT in this work was 1.92±0.11 g/cm³. This value is larger than the theoretical density of CNTs 1.5 g/cm³ reported in previous works (**Nat. Commun.**, 2014, 5, 3848) due to the existence of iron nanoparticles. The relative density for CSA-MWCNT was about 91.65±0.66%.

Related content has been added to the revised manuscript.

Reviewer's Comments 8:

If the films are crystalline and aligned, approximately how many stacked CNT layers existed in the tested samples?

Author's response:

Thank you very much for your good suggestions.

The d spacing (d-spacing) was identified to be 0.31 nm by the diffraction pattern in transmission electron microscope images in **Figure 1**, which is similar to that d-spacing reported for self-assembled single-walled CNT in the literature (**Nat. Mater.**, 2006, 5, 987-994; **Nano Lett.** 2008, 8, 1071-1075). According to the distribution of the number of walls in **Figure S1**, an average diameter (D_{CNT}) of 2.85 nm for MWCNTs was used in the calculation as shown in the following **Figure S9**. A number of CSA-MWCNT layers of 217 was obtained with the equation: number of the layers = thickness of CSA-MWCNT film / (average diameter of CSA-MWCNT + d-spacing) $\times \sin 60^\circ = 600 / (2.85 + 0.31) \times 0.866$.

Figure S9 Illustration of MWCNT packing in the CSA-MWCNT film for layer number calculation.

Reviewer's Comments 9:

The acronym “CSA” was never defined.

Author's response:

The acronym “CSA” was already defined in line 67 in the original manuscript.

Reviewer's Comments 10:

In line 132, why is a “~” symbol present when reporting conductivity, followed by 3 significant digits?

Author's response:

Thank you very much for your good suggestions. All “~” symbols have been deleted and the error bars have been added, e.g. “~9.92 MS/m” was replaced by “9.92±1.74 MS/m”.

Reviewer's Comments 11:

Line 222: a temperature gradient has units of K/m.

Author's response:

We truly appreciate your time and efforts. We have checked the manuscript carefully however we didn't find "K/m" in the original manuscript.

218 PF_{\perp} value was only half of the PF_{\parallel} (CSA-MWCNT). A thermoelectric generator
219 (TEG) was fabricated with 10 pairs of legs to demonstrate the heat to electricity
220 conversion ability of the obtained high PF_{\perp} CSA-MWCNT films (**Figure 2f**). It
221 could generate an output voltage of 1.2 mV when being held by human hands
222 at a temperature gradient of ~ 5 K (**Figure 2f**). The TEG could generate a
223 short-circuit current and an open-circuit voltage were ~ 0.4 mA and ~ 13.1 mV,
224 respectively, at a temperature gradient of 60 K. The maximum output power
225 was ~ 1.2 μ W when the loading resistance equaled to the inner resistance of 35
226 Ω for the fabricated TEG at the temperature gradient of 60 K. Detailed

Reviewer's Comments 12:

Was the Raman spectroscopy laser polarized in a specific orientation relative to the fiber orientation (Figure S2)? If yes, what was the polarization? Of no, how may the differences in Raman spectra with respect to orientation be justified?

Author's response:

We truly appreciate the reviewer's time and efforts.

A polarized laser was used in the Raman spectroscopy as shown in **Figure S3**. The parallel spectra were obtained with a polarized laser that was parallel to the winding direction of the MWCNT film. And the perpendicular spectra were obtained with a polarized laser that was perpendicular to the winding direction of the MWCNT film.

Figure S3c and S3d Illustration of the Raman spectra tests with polarized laser.

Reviewer's Comments 13:

Numerous grammatical errors are present in the manuscript. I would recommend a thorough review from a native English speaker.

Author's response:

Thank you very much for your good suggestions. The manuscript has been checked and polished carefully.

Referee 2

General comments:

The authors studied the electrical conductivity of multifunctional carbon nanotube (CNT) films synthesized by a floating chemical vapor deposition method. The synthesized CNT films were purified at high temperatures and treated with concentrated hydrochloric (HCl) acid. The authors report that the densified structure of carbon nanotubes shows high electromagnetic interference shielding efficiency, high thermoelectric power factor, and high ampacity. The authors used ChloroSulfonic Acid (CSA) treatment method. The authors also immersed the HCl-Multi Wall CNT films in CSA and heated them to 150°C in a culture dish until the added CSA was dry. This approach is a simple modification of a previously reported dipping and drying at room temperature process. The research work is nice. However, I do not see this approach as innovative, and the findings are not that promising that it can be published in nature communications. In addition to that, I have some questions for the authors.

Author's response:

Thank you very much for your time and efforts.

The entire material processing procedure in this work is unique and non-reported, in which several steps may be similar to the previously reported method. It is not just a simple modification of the dipping and drying approach. Before the dipping and drying process, the as-synthesized MWCNT film was compressed, annealed, and treated with HCl. In addition, the dipping and drying process has also been optimized. The high temperature of 150 °C was used to densify the MWCNT films. While the samples were dried at room temperature for the previously reported dipping and drying process. The material processing procedure was carefully designed and optimized.

Simply modifying the previous method will not provide high electrical and mechanical properties in MWCNT films. Even a little step of progress in the electrical and mechanical properties improvement of CNT films is not simple. Single carbon nanotubes (CNTs) have excellent electrical and mechanical properties, which have attracted much attention since it was reported in 1991. However, remaining these outstanding properties in macroscopic scale CNT-based materials is very challenging as claimed by Behabtu *et. al.* (**Science**, 2013, 339, 182). After decades of study, the electrical conductivity of CNT films was still <1 MS/m (**ACS Nano**, 2020, 14, 14134–14145), which was far away behind the theoretical value of single CNTs (>100 MS/m). In this work, the electrical conductivity was enhanced to 9.92 ± 1.74 MS/m, which resulted in the high performance of the MWCNT films in practical applications such as electromagnetic interference shielding, thermoelectrics, *etc.* This work demonstrates a new way to high-performance MWCNT films at the macroscopic scale.

Reviewer's Comments 1:

Properties of self-assembled carbon nanotubes (works is for Single Wall CNT, not for Multi-Wall CNT, however) are presented in Nano Lett. 19, 5, 3131 – 3137 (2019) by Roberts et al. analyzing dielectric functions. The dielectric function and conductivity are related to each other through a well-known equation. Smaller ϵ implies larger σ . How much do your findings agree?

Author's response:

Thank you very much for your time and efforts. We sincerely appreciate it. The dielectric function and the electrical conductivity may be related to each other for MWCNT films. However, we couldn't find the equipment to experiment with dielectric function analyses, which should be out of the research scope of this work.

The purpose of this work is to report a material processing procedure for MWCNT films with high electrical and mechanical properties at the macroscopic scale. These films exhibit high performance in practical applications such as electromagnetic interference shielding, thermoelectrics, *etc.* It has been demonstrated that the electrical conductivity of MWCNT film was related to the alignment, packing density, doping level and purity of MWCNTs, *etc.* The relationship between the dielectric function and the electrical conductivity of MWCNT films is important, which will be studied in our next work.

Reviewer's Comments 2:

In the literature, it is also noted that the dielectric response and the electrical conductivity of CNT films are highly anisotropic, and low conductivity in a particular direction is due to the depolarization effect [for example, Bondarev et al. Phys. Rev. Applied 15, 034001 (2021)]. The authors are missing a core concept.

Author's response:

Thank you very much for your good suggestions.

The purpose of this work is to report a material processing procedure for MWCNT films at a macroscopic scale with high electrical and mechanical properties. As suggested by Fukuhara *et. al.*, (**Appl. Phys. Lett.**, 2018, 113, 243105) and Sun *et. al.*, (**Adv. Funct. Mater.**, 2022, 32, 2203080), the anisotropic electrical conductivity of MWCNT film at the macroscopic scale was explained well with Mott's theory. The discussion on the anisotropic electrical conductivity has been provided in the original manuscript. Further explanation of the highly anisotropic electrical conductivity is out of the research slope which will not change the main conclusion of this work as well.

Reviewer's Comments 3:

The authors have used the term MWCNT in many places in the manuscript. However, in several places, the authors only write CNT. Which CNT authors are talking about in these places? Are these CNT Single Wall CNT or Multi-Wall CNT?

Author's response:

Thank you very much for your time and efforts. The CNTs were synthesized with a floating catalytic chemical vapor deposition method, which often provided a mixture of single-walled CNTs and multi-walled CNTs. The distribution of the number of walls per CNT was shown in **Figure S1** in the original supporting information. It has been demonstrated that the majority CNTs were multi-walled CNTs in this work.

Thanks to the reviewer's good suggestion. The related "CNT" has been replaced with "MWCNT" in the revised manuscript to avoid misunderstanding.

Reviewer's Comments 4:

The authors mentioned that the winding drum to collect CNT was a diameter of 10 cm. What about the size of CNT they used and the MWCNT they prepared?

Author's response:

We truly appreciate the reviewer's time and efforts.

The optical image of the pristine-MWCNT film has been added in **Figure S1k** in the revised supporting information. Typically, the film has a length of 25 cm and a width of 28 cm. The sizes of MWCNT films used in this work have been summarized in **Table S7**.

Figure S1k optical image of the synthesized pristine-MWCNT film

Table S7 Summary of the sizes of MWCNT films used in this work

Sample	Size Length x width (cm)
SEM	1x1
Raman	1x1
XRD	1x1
Thermoelectric	2x0.5
Electromagnetic interference	2.5x2.5
Current density	3x0.1
Strength	1x0.5

Related content has been added to the revised manuscript.

Reviewer's Comments 5:

It is said that collecting CNT in winding drum pristine-MWCNT was prepared. I am bothered with the word pristine as there is no clear figure of the sample showing MWCNT, and the authors have a very little discussion about this.

Author's response:

The synthesis of the pristine-MWCNT was described in the section “formation of CNT film” in the original manuscript. An optical image has been added to the supporting information in **Figure S1k** as shown below. A detailed description has been added to the revised manuscript as shown below:

“When the collection was complete after 180 min, the film was peeled off from the drum. Then it was compressed for 5 min with a pressing machine (Kejing MSK-2150 China) at a force of 4 tons to make the surface of the MWCNT films smooth. After that, the obtained pristine MWCNT films were dried in vacuum at a temperature of 50 °C and kept in the glove box before used.”

Figure S1k optical image of the synthesized pristine-MWCNT film

Reviewer's Comments 6:

The literature is rich in the study of thickness-dependent properties of CNT film. How about the thickness of your film? How the conductivity of your film depends on film thickness?

Author's response:

We truly appreciate the reviewer's time and efforts.

The following **Table S1** listed the electrical conductivities and thicknesses in the parallel direction of CNT films. The thicknesses of pristine-MWCNT, annealed-MWCNT, HCl-MWCNT and CSA-MWCNT were $8.34 \pm 0.92 \mu\text{m}$, $7.07 \pm 0.35 \mu\text{m}$, $6.64 \pm 0.19 \mu\text{m}$, $0.64 \pm 0.05 \mu\text{m}$, respectively, as shown in **Figure S7**. The purpose of this work is to report a material processing procedure for MWCNT films at a macroscopic scale with high electrical and mechanical properties. In this work, it was demonstrated that the electrical conductivity of MWCNT film was related to the alignment, packing density, doping level and purity of MWCNTs, *etc.* The thickness-dependent properties of CNT film should be out of the research scope, which will be studied in our next work.

Table S1 The electrical conductivities and thicknesses of CNT films

Sample	Thickness (μm)	Electrical conductivity σ_{\parallel} (MS/m)	Electrical conductivity σ_{\perp} (MS/m)
Pristine-MWCNT	8.34 ± 0.92	0.18 ± 0.03	0.04 ± 0.006
Annealed-MWCNT	7.07 ± 0.35	0.49 ± 0.023	0.11 ± 0.006
HCl-MWCNT	6.64 ± 0.19	2.33 ± 0.24	0.56 ± 0.07
CSA-MWCNT	0.64 ± 0.05	9.92 ± 1.75	3.88 ± 0.42

Referee 3

General comments:

The authors report on a dry-drawn, and chemically-treated MWCNT sheet showing multi-functionalities including high EMI shielding, thermoelectric power factor, current ampacity, and mechanical strength. This reviewer considers a part of the properties demonstrated here (e.g. ultrahigh electrical conductivity) is interesting as a carbon-based nanomaterial while its comparison with standard materials such as silver films for EMI, Bi₂Te₃ for thermoelectrics, and carbon fibers/CFRP for mechanics is fair. Several additional points are noted below.

Author's response:

Thank you very much for your time and efforts. We sincerely appreciate your support.

Single carbon nanotubes (CNTs) have excellent electrical and mechanical properties, which have attracted much attention since it was reported in 1991. However, remaining these outstanding properties in macroscopic scale CNT-based materials is very challenging as claimed by Behabtu *et. al.* (**Science**, 2013, 339, 182). After decades of study, the electrical conductivity of CNT films was still <1 MS/m (**ACS Nano**, 2020, 14, 14134–14145), which was far away behind the theoretical value of single CNTs (>100 MS/m). In this work, the electrical conductivity was enhanced to 9.92 ± 1.74 MS/m, which resulted in the high performance of the MWCNT films in practical applications such as electromagnetic interference shielding, thermoelectrics, *etc.* This work demonstrates a way to high-performance MWCNT films at the macroscopic scale.

Reviewer's Comments 1:

The authors claimed “light” . They should note the quantitative density (g/cc) of each material.

Author's response:

Thank you very much for your good suggestion.

The density of pristine-MWCNT was 0.49 ± 0.11 g/cm³, which increased to 1.92 ± 0.11 g/cm³ for CSA-MWCNT. The density values are comparable to that of light-weight CNT films in the range of 1.5-2 g/cm³ reported in the literature (1.5 g/cm³, **Nat. Commun.**, 2014, 5, 3848; 1.8 g/cm³, **Nat. Commun.**, 2014, 5, 3848; 2 g/cm³ **Adv. Funct. Mater.**, 2022, 2203080)

Related content has been added to the revised manuscript.

Reviewer's Comments 2:

Thermal conductivity, along with power factor, is important for showing thermoelectric efficiency.

Author's response:

Thank you very much for your good suggestions.

Additional experiments have been performed to get the thermal conductivities of pristine-MWCNT and CSA-MWCNT in the in-plane direction by a laser flash method with NETZSCH LFA-467 (German). The results were listed in **Table S5** as shown below. Three samples were tested for each MWCNT film. The average thermal conductivities for pristine-MWCNT and CSA-MWCNT were 21.93 ± 0.10 and 45.90 ± 0.10 W/m-K, respectively. It is interesting to note that the figure-of-merit of CSA-MWCNT films has increased by 3 times as compared to that of pristine-MWCNT films. The obtained materials are promising for active cooling (Peltier effect) as suggested by Kono *et. al.* in the literature, where high power factor and high thermal conductivity are required (**Nat. Commun.**, 2021, 12, 4931; **ACS Energy Lett.**, 2021, 6, 4355).

Table S5 The measured thermal conductivities at room temperature.

Sample	Thermal conductivity (W/m-K)				ZT
	1	2	3	Average	
Pristine-MWCNT	21.8	22.0	22.0	21.93 ± 0.10	0.0143 ± 0.0017
CSA-MWCNT	45.9	45.8	46.0	45.90 ± 0.10	0.0304 ± 0.0053

Related content has been added to the revised manuscript.

Reviewer's Comments 3:

This reviewer did not understand the mechanism of CSA-driven densification. The authors should explain it around ~line 122 after the introduction of reference 25.

Author's response:

Thank you very much for your good suggestions. A detailed explanation of the mechanism has been added in the revised manuscript.

The CSA-driven densification method was first reported by Behabtu *et. al.* (**Science** 2013, 339, 182-186) for light-weight, strong and highly conductive carbon nanotube fibers. Later on, the same group used this method to further improve the electrical and mechanical properties of carbon nanotube fibers (reference 25: **Carbon**, 2021, 171, 689-694). The reason for the densification of carbon nanotube fiber was attributed to the increased interaction between two carbon nanotubes after CSA

treatment. After being treated with CSA acid, individual CNTs were charged, which led to many negative and positive sites on the CNTs due to the charge separation. These charged sites would increase the interaction between two CNTs through the Coulomb force. In the meanwhile, the van der Waals force between two CNTs also played an important role in the interaction between two CNTs. The Coulomb force together with the van der Waals force thus led to the high strength of the MWCNT film. A similar mechanism has been proposed in the literature (**ACS Nano**, 2020, 14, 14134–14145).

Related content has been added in the revised manuscript near line 122.

Reviewer's Comments 4:

Long-term stability data (Figure 3f) show the gradual de-doping, where electrical conductivity decreased and the Seebeck coefficient slightly increase. Additionally, CAS seems to show moderate volatility, potentially leading to gradual de-doping. As such, this reviewer did not approve for the statement of line 286-288 “The results ... with stable electrical and thermoelectric properties.

Author's response:

Thank you very much for your good suggestion. We are sorry that the statement is not rigorous.

The statement on line286-288 “ The results ... with stable electrical and thermoelectric properties” has been replaced with “The results demonstrated that the CSA-MWCNT films possessed high mechanical properties and their PF even increased by 8% for a long time of 40 days due to the increase of Seebeck coefficient of CSA-MWCNT film caused by the de-doping effects in the air. ”

Reviewer's Comments 5:

Mechanical properties for perpendicular direction against alignment should be noted.

Author's response:

Thank you very much for your good suggestions.

Addition experiments have been performed to get the mechanical properties of CSA-MWCNT films in the direction perpendicular to the winding direction. **Figure S13** showed that the tensile strength of CSA-MWCNT in the direction perpendicular to the winding direction increased to 0.218 GPa which was about 16 times higher than that of pristine-MWCNT films. The increased tensile strength in both the parallel and the perpendicular direction to the winding direction demonstrated the increase of interactions between two carbon nanotubes in the CSA-MWCNT films.

Figure S13 The tensile strength of the pristine-MWCNT film and CSA-MWCNT film in the direction perpendicular to the winding direction.

Related content has been added to the revised manuscript.

Reviewer's Comments 6:

The ampacity presented here was much lower than single tubes and SWCNT-Cu composites, as noted in the manuscript. Could you describe its reason in the manuscript?

Author's response:

Thank you very much for your good suggestions. We sincerely appreciate it.

The lower ampacity of CSA-MWCNT film is attributed to the higher electrical resistivity and lower thermal conductivity derived from the non-uniformity of the film compared to single tubes and SWCNT-Cu composites. Because CSA-MWCNT films contain irregular iron nanoparticles, which leads to lower electrical conductivity and thermal conductivity. As suggested by Behabtu *et. al.* (**Science**, 2013, 339, 182), the higher electrical and thermal conductivity in single tubes and SWCNT-Cu reduced the material temperature by generating less Joule heating and by dissipating the heat efficiently, preventing the broken of the sample.

Related content has been added to the revised manuscript.

Reviewer's Comments 7:

Several mistypes should be checked. For example,

Line 205: the Wiedemann-Franz law => (other physics law?)

* The W-F law explains the relationship between the electrical and thermal conductivity of ideal metals. This is not associated with the Seebeck coefficient.

Line 222: generat => generate

Author's response:

We sincerely appreciate your time and efforts. The manuscript has been checked carefully. All errors and typos should have been corrected in the revised manuscript.

Line 205: The "Wiedemann-Franz law" has been replaced with "Pisarenko relation".

Line 222: The "generat" has been replaced with "generate".

REVIEWER COMMENTS

Reviewer #1 (Remarks to the Author):

The authors have addressed well the reviewer concerns, and the manuscript has benefited from the process.

One remaining concern is response comment number 11. The intent of the comment was to convey to the authors that a temperature gradient has units of K/m. The manuscript reports a temperature gradient in terms of K. I believe that the authors are referring to a temperature difference rather than a temperature gradient. A temperature difference is perfectly acceptable, but accurate terminology is important.

Reviewer #2 (Remarks to the Author):

I thank the authors for their answers and explanations to my questions and concerns. The authors addressed most of my concerns, although some answers are not entirely satisfactory but understandable. It would be nice if the authors put extra effort into completing the task now than answering that they will perform this task in the upcoming research work.

Reviewer #3 (Remarks to the Author):

The authors have revised their manuscript according to the referees' comments. After carefully reading the authors' revision, however, I could still find no significant points outperforming the conventional materials in each application field, which has been suggested at the 1st round reviewing. The data collection is not suitable for the publication.

Rebuttal

Manuscript ID: NCOMMS-22-30697-A

Title: Acid enhanced zipping effect to densify MWCNT packing for strong, light, multifunctional MWCNT films with ultra-high electrical conductivity

Referee 1

General comments:

The authors have addressed well the reviewer concerns, and the manuscript has benefited from the process. One remaining concern is response comment number 11. The intent of the comment was to convey to the authors that a temperature gradient has units of K/m. The manuscript reports a temperature gradient in terms of K. I believe that the authors are referring to a temperature difference rather than a temperature gradient. A temperature difference is perfectly acceptable, but accurate terminology is important.

Authors' response:

Thank you very much for your time and efforts. We sincerely appreciate it.

We agree with the reviewer that accurate terminology is important. Thanks a lot to the reviewer for your great scientific attitude. The “temperature gradient” has been replaced with “temperature difference” in the revised manuscript.

Thanks again for helping us to improve the quality of the manuscript.

Referee 2

General comments:

I thank the authors for their answers and explanations to my questions and concerns. The authors addressed most of my concerns, although some answers are not entirely satisfactory but understandable. It would be nice if the authors put extra effort into completing the task now than answering that they will perform this task in the upcoming research work.

Authors' response:

Thank you very much for your support and insightful comments. We would be glad to complete all the tasks suggested by the reviewer. Additional experiments have been performed to show the relationship between the dielectric function and the electrical conductivity as well as the relationship between the thickness and the electrical conductivity.

The dielectric functions of the CSA-MWCNT film were measured by an ellipsometer (HORIBA UVISEL PLUS, Japan). **Figure S11** showed that the $\text{Im}(\epsilon)$ in the parallel direction was larger than that in the perpendicular direction. The result is similar to that reported in previous work mentioned by the reviewer (**Nano Lett.**, 2019, 5, 3131-3137) in the 1st round of reviewing. Here, larger $\text{Im}(\epsilon)$ implied larger sigma σ . It should be noted that the dielectric function may be affected by many factors such as the morphology, the porosity and the impurities in the CSA-MWCNT films, which often do not show a linear relationship to the electrical conductivity.

Figure S11 Dielectric functions of the CSA-MWCNT film.

The thickness dependent electrical conductivities of both pristine-MWCNT and CSA-MWCNT films at the parallel direction (σ_{\parallel}) are shown in **Figure S9**. MWCNT films with different thicknesses were obtained by changing the collection time. The

thicknesses of the films were confirmed by the scanning electron microscope (SEM) images of the cross-section of MWCNT films (**Figure S10**). **Figure S9** showed that the σ_{\parallel} of both pristine-MWCNT and CSA-MWCNT films increased with the decrease of the film thickness. The results were consistent with that reported in previous works for CNTs (**Appl. Phys. Lett.**, 2009, 94, 012904).

Figure S9 a) The σ_{\parallel} of the pristine-MWCNT film as a function of the thickness. b) The σ_{\parallel} of the CSA-MWCNT film as a function of the thickness.

Figure S10 Scanning electron microscope (SEM) images of the cross-section of pristine-MWCNT films (a-d) and CSA-MWCNT films (e-h).

Referee 3

General comments:

The authors have revised their manuscript according to the referees' comments. After carefully reading the authors' revision, however, I could still find no significant points outperforming the conventional materials in each application field, which has been suggested at the 1st round reviewing. The data collection is not suitable for the publication.

Authors' response:

Thank you very much for your time and efforts. We sincerely appreciate it. We hope that the reviewer could understand that this manuscript has explored a way to high-performance multifunctional materials at a macroscopic scale with the widely studied carbon nanotubes. The achieved chlorosulfonic acid treated multi-walled carbon nanotube (CSA-MWCNT) film possesses high properties comparable to (or even higher than) that of the state-of-the-art single-functional conventional materials mentioned by the reviewer.

It will be more clear to compare the properties with a radar plot as shown in **Figure S17**. The related data are listed in **Table S4**, **S7** and **S8**. CSA-MWCNT film possesses high specific electrical conductivity of $5.26 \times 10^4 \text{ Scm}^2/\text{g}$ comparable to that of silver films ($5.72 \times 10^4 \text{ Scm}^2/\text{g}$, **NPG Asia Mater.**, 2018, 10, 749) and much higher specific shielding efficiency of $85.5 \text{ dB}/\mu\text{m}$ than that of silver films ($5.85 \text{ dB}/\mu\text{m}$, **NPG Asia Mater.**, 2018, 10, 749). It also possesses a high power factor ($4660 \mu\text{W}/\text{m}\cdot\text{K}^2$) close to that of state-of-the-art Bi_2Te_3 films ($4700 \mu\text{W}/\text{m}\cdot\text{K}^2$, **ACS Nano**, 2021, 15, 5706) and good mechanical properties ($1.15 \text{ N}/\text{tex}$) approaching to that of carbon fibers ($4.08 \text{ N}/\text{tex}$, **Nat Commun.**, 2019, 10, 2962).

Figure S17 A radar plot showing a comparison of comprehensive performance covering specific shielding, specific electrical conductivity, power factor, 1/density and specific strength.

In practical applications, a good material is often expected to be a multifunctional material other than a single-functional material with exceptional properties in only one field (**Adv. Mater.**, 2022, 34, 2107262; **Adv. Mater.**, 2022, 34, 2110406; **Nat. Commun.**, 2021, 12, 1416). Multifunctional materials possessing lightweight, high electrical conductivity, and high mechanical properties are promising for many applications such as electromagnetic interference shielding of electronic devices, lightning strike protection of aircraft (**Funct. Compos. Struct.**, 2020, 2, 022002) and flexible thermoelectric devices (**Adv. Mater.**, 2018, 30, 1704386). However, it is challenging to combine these properties simultaneously in one material as reported by Prof. Yu (**Nano Lett.**, 2021, 21, 2532–2537) and Prof. Young (**Science**, 2013, 339, 182).

Single CNT has a relatively low density of $\sim 1.6 \text{ g/cm}^3$, which possesses excellent electrical and mechanical properties as well. However, CNT films often have low electrical and mechanical properties far away behind single CNT due to the handling challenges in material processing (**Science**, 2013, 339, 182). Big progress of CNT films in terms of the electrical conductivity (**Figure 1d**), the electromagnetic interference shielding efficiency (**Figure 2b**) and the thermoelectric power factor (**Figure 2e**), *etc.*, has been achieved in this paper, which makes the CSA-MWCNT film a promising multifunctional material. This work may promote the application of CNTs in the fields of electromagnetic interference shielding and flexible thermoelectrics, *etc.*

Related content has been added to the revised manuscript.

Table S4 $PF_{||}$ (CSA-MWCNT) was compared with the PF values of organic and inorganic films in the literature

	Sample	Electrical conductivity (MS/m)	Seebeck coefficient (μ V/K)	Power factor (μ W/m-K ²)	Date	
	CSA-MWCNT	~10	~23	~4660	This work 2022	
Organic materials	Oxygen plasma treatment of few-layer graphene	~0.07	~700	4500	2011 ⁵⁷	
	a-CNT web/BV PANi/graphene-PEDOT:PSS/DWNT-PEDOT:PSS	~0.22	-116	3103	2017 ⁵⁸	
	PANi/graphene/PANi/DWCNT	~0.11	130	1825	2015 ⁶⁰	
	MWCNT/TCNQ	0.89	45	1800	2022 ⁹	
	PEDOT:PSS/CuCl ₂	5.2x10 ⁻⁶	-18200	1700	2020 ⁶¹	
	SWCNT/PEI	~0.36	-64	1500	2017 ⁶²	
	TDAE-PEDOT/CNT	7.3x10 ⁻⁴	-1200	1050	2015 ⁶³	
	MWCNT/PEI	0.50	-45	1000		
	SWCNT/polystyrene	~0.21	61	789	2020 ⁶⁴	
	MWCNT/[HMIM][BF ₄]	~0.11	~80	762	2022 ⁶⁵	
	SWCNT/FcMA	~0.27	-46.07	567.54	2021 ⁶⁶	
	TPETPA/SWCNT	~0.04	123.2	539.8	2018 ⁶⁷	
	PEDOT:PSS/SWCNT	0.17	55.6	526	2019 ⁶⁸	
	poly(metaTFSI)/SWCNT	0.1	70	490	2017 ⁶⁹	
	PEDOT:PSS/PSSH(PSSNa) coating after H ₂ SO ₄	0.21	43.5	401	2018 ⁷⁰	
	SWCNT/PANi	~0.24	39.2	362	2020 ⁷¹	
	PEDOT-PF ₆ /SWCNT	0.36	31.1	350	2019 ⁷²	
	PEDOT:PSS/H ₂ SO ₄ , NaOH drop	~0.22	39.2	334	2017 ⁷³	
	Inorganic materials	Bi ₂ Te ₃	~0.15	-180	4700	2021 ⁷⁴
		single-crystalline Bi ₂ Te ₃	~0.13	164	3400	2021 ⁷⁵
Bi ₂ Te ₃		~0.07	222	3373	2022 ⁷⁶	
Bi ₂ Te ₃		~0.08	-180	2500	2020 ⁷⁷	
Bi ₂ Te ₃		~0.04	-198.9	1490	2021 ⁷⁸	
Bi ₂ Te ₃		~0.05	165	1460	2022 ⁷⁹	
Bi ₂ Te ₃		~0.04	184.2	1250	2021 ⁷⁸	
Bi ₂ Te ₃ pellet		~0.07	-133	1289	2019 ⁸⁰	

Table S7 Comparison of the specific shielding efficiency, the density, the specific strength and the specific electrical conductivity of silver film, Bi₂Te₃ film and CSA-MWCNT film.

Sample	Power factor (μ W/m-K ²)	SSE (dB/ μ m)	Density (g/cm ³)	Specific strength (N/tex)	Specific electrical conductivity (Scm ² /g)
Ag film	~266 ⁸⁷	~5.84 ⁴⁴	10.49 ⁴⁴	~0.016	~5.72x10 ⁴
Bi ₂ Te ₃ film	4700 ⁷⁴	~0.027 ⁸⁸	7.7	~0.922 ⁸⁹	~194.8 ⁹⁰
This work	~4660	85.5	~1.9	~1.15	5.26 x10 ⁴

Table S8 Comparison of the tensile strength of the CSA-MWCNT film with CNT based fibers in the literature.

	Sample	Specific tensile strength (N/tex)	Tensile strength (GPa)	Density (g/cm ³)	Date
	CSA-MWCNT film	1.15	2.2	~1.9	This work
Fiber	CNT	4.08	4.48	1.1	2019 ⁹¹
	CNT	3.84	6.57	1.71	2022 ⁹²
	Cross-linked CNT	3.7	1.7	0.45	2017 ⁹³
	G-CNT	3.00	6.05	2.01	2022 ⁹⁴
	PI-CNT	2.99	6.21	1.74	2022 ⁹⁵
	CNT	2.55	5.02	1.97	2022 ⁹⁶
	CNT	2.1	4.2	0.51	2021 ⁹⁷
	DWCNT	1.6	2.4	1.5	2017 ⁹⁸
	CNT/I ₂	0.97	1.35	1.4	2013 ⁸⁴
	CNT	~0.91	-	-	2021 ⁹⁹
	CNT	0.89	1.33	1.49	2022 ¹⁰⁰
	CNT/py-PDA/C	0.79	0.727	0.92	2019 ¹⁰¹

REVIEWERS' COMMENTS

Reviewer #1 (Remarks to the Author):

The authors have done an acceptable job addressing the reviewer comments. The remaining concern is advanced by reviewer 3 - the overall significance of the observed properties. The authors have supplied an argument regarding multifunctionality that is reasonable. In my estimation, the manuscript is suitable for publication.

Reviewer #2 (Remarks to the Author):

The authors have adequately addressed my concerns and improvised the manuscript significantly. The manuscript is worth publishing now. There are a few typographical errors, however. I hope the authors will take care of these typos.

Reviewer #3 (Remarks to the Author):

I thank the authors for addressing my concern, and new Figure S17 is very impressive. I expect their future work on thermal application published elsewhere, since huge electrical conductivity would be associated with corresponding thermal conductivity.

Response Letter to Referees: 3rd round

Reviewer Comments to Author:

Reviewer #1 (Remarks to the Author):

The authors have done an acceptable job addressing the reviewer comments. The remaining concern is advanced by reviewer 3 - the overall significance of the observed properties. The authors have supplied an argument regarding multifunctionality that is reasonable. In my estimation, the manuscript is suitable for publication.

Authors' response:

Many thanks to the reviewer. We sincerely appreciate your time and efforts on helping us improving the quality of this manuscript.

Reviewer #2 (Remarks to the Author):

The authors have adequately addressed my concerns and improvised the manuscript significantly. The manuscript is worth publishing now. There are a few typographical errors, however. I hope the authors will take care of these typos.

Authors' response:

Thank you very much for your time and efforts. We sincerely appreciate your kindness support. The manuscript has been revised carefully. All the errors and typos should have been corrected in the manuscript.

Reviewer #3 (Remarks to the Author):

I thank the authors for addressing my concern, and new Figure S17 is very impressive. I expect their future work on thermal application published elsewhere, since huge electrical conductivity would be associated with corresponding thermal conductivity.

Authors' response:

Thank you very much for your time and efforts. We sincerely appreciate your great suggestions. We agree with the reviewer that the thermal conductivity will increase with the electrical conductivity. The thermal conductivity of the MWCNT films will be studied in our future work.